# Frequency, Correlates, and Symptom Severity of Eating Disorders Among College Students in Mexico

**DOI:** 10.3390/ijerph22121797

**Published:** 2025-11-27

**Authors:** María Alonso-Catalán, Silvia A. Tafoya, Rosalia Vazquez-Arevalo, María Luisa Ávila-Escalante, María Teresa Tusié-Luna, Lidia Moreno-Macías, Hortensia Moreno-Macías, Julián Corral-Aguilar, Mónica Aburto-Arciniega, Santiago Silva-Avalos, Diego Armando Coronel-Manzo, Benjamín Guerrero-López, Claudia Díaz-Olavarrieta

**Affiliations:** 1Department of Psychiatry and Mental Health, Faculty of Medicine, National Autonomous University of Mexico, 3000 Ave. Universidad, Copilco Universidad, Coyoacán, Ciudad Universitaria, Mexico City 04510, Mexico; mc20aocm5338@facmed.unam.mx (M.A.-C.); stafoya@unam.mx (S.A.T.); diego.coronel@facmed.unam.mx (D.A.C.-M.); dr.bguerrero@unam.mx (B.G.-L.); 2Facultad de Estudios Superiores (F.E.S.) Iztacala, National Autonomous University of Mexico, Av. de los Barrios No. 1, Los Reyes Iztacala, Tlalnepantla 54090, Mexico; rvamap@unam.mx; 3Faculty of Medicine, National Autonomous University of Yucatan, Calle 60 No. 491-A x 57 y 59, Centro, Mérida 97000, Mexico; marialuisa.avila@correo.uady.mx (M.L.Á.-E.); lmmacias@correo.uady.mx (L.M.-M.); 4Unidad de Biología Molecular y Medicina Genómica, Instituto de Investigaciones Biomédicas de la UNAM, Mexico City 70228, Mexico; mttusie@iibiomedicas.unam.mx; 5Instituto Nacional de Ciencias Médicas y Nutrición Salvador Zubirán, Vasco de Quiroga 15, Belisario Domínguez Sección VXI, Tlalpan, Mexico City 14080, Mexico; 6Division of Social Sciences and Humanities, Universidad Autónoma Metropolitana Iztapalapa, San Rafael Atlixco No. 186, Leyes de Reforma 1ra Sección, Iztapalapa, Mexico City 09310, Mexico; hmm@xanum.uam.mx; 7Hospital de Especialidades, Centro Médico Nacional Siglo XXI, Mexican Social Security Institute (IMSS), Av. Cuauhtémoc 330, Doctores, Cuauhtémoc, Mexico City 06720, Mexico; juliancorral@comunidad.unam.mx; 8Faculty of Medicine, National Autonomous University of Mexico, 3000 Avenida Universidad, Copilco Universidad, Coyoacán, Ciudad Universitaria, Mexico City 04510, Mexico; maburto@unam.mx; 9Department of Mathematics, Division of Actuarial Science, Mathematics and Statistics, Instituto Tecnológico Autónomo de México, Río Hondo No. 1, Progreso Tizapán, Álvaro Obregón, Mexico City 01080, Mexico; silvaavalos.santiago@bcg.com

**Keywords:** eating disorders, body dissatisfaction, college students

## Abstract

The prevalence of eating disorders (ED) has grown in recent decades, associated with sociocultural pressures that partly stem from body stereotypes and increased stress in daily life. However, even though college constitutes a high-risk life stage, this group has received little attention, leaving a significant gap in prevention and early intervention. This cross-sectional and comparative study included 610 Mexican college students aged 18–30 enrolled in two public universities in Mexico: one sample comprised undergraduate students from Yucatan’s UADY (n = 457), and another one from Mexico City’s UNAM (n = 153). Screening tools included the EAT-26, BES, BSQ, DMS, HADS and AUDIT. We determined the frequency of ED symptomatology across both universities. Linear and multiple logistic regressions were conducted to identify factors associated with ED symptomatology. Overall, 19.8% of participants displayed significant ED symptomatology (UADY 15.5%, UNAM 32.7%). Significant ED symptomatology was more common among women (31.8%) than men (11.7%). Non-cisgender students (27.8%) showed a higher frequency than cisgender women (24.7%) and cisgender men (11.2%). High BD was the strongest predictor of significant ED symptomatology (OR = 13.35; 95% CI 6.79–26.26), followed by symptoms of anxiety (OR = 1.11, 95% CI 1.03–1.19). Our findings highlight the need for early screening and targeted interventions among college students in Mexico.

## 1. Introduction

### 1.1. Eating Disorders (ED) and Body Dissatisfaction (BD)

Eating disorders (ED) are debilitating psychiatric conditions that include anorexia nervosa (AN), bulimia nervosa (BN) and binge eating disorder (BED). EDs are characterized by restrictive calorie intake, limitations in the quantity and/or type of food, binge eating, and compensatory behaviors that lead to malnutrition and exaggerated concern about weight and body image [1]; they are intertwined with multifactorial genetic, psychological, and sociocultural etiological factors [2]. AN, BN, and BED primarily affect adolescents and young adults, with a higher prevalence among women [3]. Empirical data on the incidence and prevalence of the less common DSM-V ED subtypes (i.e., pica, rumination disorder, avoidant/restrictive food intake disorder) remains sparse, largely because conflicting operational criteria make comparisons across studies difficult. Body dissatisfaction (BD) is a negative self-assessment of one’s physical appearance, which can lead to both physical and mental health consequences [4]. Similar to ED, the prevalence of BD is higher among young women and is commonly associated with low self-esteem and symptoms of anxiety and depression [4]. Prior research has identified BD as a risk factor for an ED [5,6,7,8].

#### 1.1.1. Global Prevalence

Women in Western countries report a high prevalence of ED often linked with BD and mood disorders such as depression and anxiety [9,10]. A systematic review (2000–2018) estimated the lifetime prevalence of ED at 8.4% in women (range: 3.3–18.6%) and 2.2% in men (range: 0.8–6.5%) [11]. These findings align with data from the 2019 Global Burden of Disease Study, which reported that mental disorders affected over 970 million people globally, with anxiety and depressive disorders being the most common. Notably, these mental health conditions—together with ED—were more prevalent among women [12]. The burden of mental health issues in young adults is further reflected in substance use patterns. According to the 2023 National Survey on Drug Use and Health in the United States, 49.6% of young adults aged 18 to 25 reported alcohol consumption in the past month—defined as drinking any alcoholic beverage, with a standard drink containing 0.6 fl oz of pure alcohol. In the same period, 6.8% of full-time college students reported heavy drinking—defined as five or more drinks for males on a single occasion or fifteen or more drinks per week, and as four or more drinks for females on a single occasion or eight or more drinks per week [13,14].

#### 1.1.2. Eating Disorders in Mexico

Research on ED in Mexico is relatively new compared to other middle-income countries [15,16]; however it is gaining increasing importance as recent findings reveal the high prevalence of ED among young adults. González-Alvarado et al. (2024) conducted a cross-sectional study that included 395 college students and found that 37.3% of women and 32.6% of men were at risk of developing an ED [17]. Risky Eating Behaviors (REB) were addressed in the country’s most ambitious health survey, the National Survey on Health and Nutrition (ENSANUT for its initials in Spanish). The 2018–2019 results reported a prevalence of 1.9% in women and 0.7% in men. The 2022 wave revealed a significant increase, with 6.6% of adolescents (both genders) displaying REB- 5% with moderate risk and 1.6% at high risk [9]. A 2014 cross-sectional study also found an elevated prevalence of REB among medical students (8.6%) higher among male participants (9.4% vs. 7.4%) [18]. Villalobos-Hernández (2023) reported that between 5% and 18% of female participants exhibited REB, including heightened concern about weight gain and the use of purging and restrictive eating behaviors [9]. While body weight overestimation declined among men, it increased among women, who also showed a higher prevalence of binge eating [9]. A 2008 study of nutrition students in Yucatán found a prevalence of ED of 9% among women and 13% among men. In the same study, 16% of female participants and 22% of male participants also reported dissatisfaction with their body shape. These findings are notable due to the higher prevalence observed among men compared with women—a result that contradicts most prior research on the subject [19]. Yucatán’s ED prevalence may be associated with its mental health indicators, as the state has a suicide rate roughly double that of other Mexican states. A cross-sectional study (2022) reported associations between depression with anxiety, suicidality and overall well-being [20]. Despite these concerning mental health indicators, research on Yucatán’s ED situation remains scarce.

In Mexico’s history, ED have been reported as more prevalent among women than men [21,22]. Nevertheless, recent studies suggest that classic forms of ED—particularly AN and BN—may be underdiagnosed in men [23]. Only in recent years has there been increased attention to studying sex differences in the onset, comorbidities, outcomes and symptomatology of ED [24]. While women’s ideal body is based on a thinness and thin-idealization [25], men’s body ideals focus on muscularity and low body fat, often resulting in strict dietary regimes and intense physical training [24,26]. Men more frequently express the desire to gain more weight, in contrast to women, who tend to pursue weight loss and consequently exhibit higher rates of purging behaviors such as self-induced vomiting and the misuse of diuretics or laxatives [27]. The male ideal body is partly influenced by cultural norms of hegemonic masculinity that valorize strength, invulnerability and rejection of femininity [28]. Consistent with this, men exhibit higher risk of developing “muscle dysmorphia” or “bigorexia” than women [24,27]. Additionally, men are less likely to seek mental health services, a pattern attributed in part to stigma and gendered expectations surrounding help-seeking behaviors [29,30]. When men do engage with clinical services, their symptoms are more likely to be overlooked or misclassified due to persistent misconceptions that ED primarily affect women [24]. Despite the substantial prevalence of ED in Mexico, it is estimated that fewer than 10% of individuals— regardless of sex—receive appropriate treatment [31].

Importantly, gender identity and sexual orientation further shape ED vulnerability. Lesbian, gay, bisexual and transgender (LGBT+) individuals experience a higher prevalence of ED, likely due to experiencing greater stress caused by social stigma and prejudice [32]. In Mexico, sexual minority men are at greater risk of exhibiting ED symptomatology [33]. According to Cervantes-Luna et al. (2019), homosexual and bisexual men present with more BD compared to heterosexual men and, similar to women, show greater internalization of thinness ideals and weight-loss behavior [34].

In 2024, an estimate of 5.07 million people used social media [35]. Mexico has above 47 million Instagram users who spend an average of 6.5 h a month on the platform. About 66% of Instagram users in Mexico are between 18 and 34 years old [36]. Social media misuse has been identified as a significant contributing factor to the development of ED symptomatology [37], likely due to its association with a social comparison tendencies [38,39]. Instagram and other visually oriented platforms are associated with higher rates of appearance comparison and body dissatisfaction compared with non-visual platforms. Repeated exposure to idealized body images—typically thin for women and muscular for men—can foster internalization of these body ideals and contribute to elevated BD [38].

#### 1.1.3. Comorbidities of ED

ED are associated with adverse physical and mental health consequences often linked to a mortality rate over than three times higher than the general population [40]. Over half of people with an ED have a comorbid depressive or anxiety condition [41]. In addition to mental health conditions, people with ED also face higher risks of substance abuse and suicide [42]. Studies show that women, more than men, experience greater mental health outcomes associated with binge behaviors (food), body mass index (BMI), and risky eating patterns [43]. Food and alcohol binges may share several important characteristics, including attempts to regulate negative mood states, such as depression or anxiety. The traits found in both types of binges include: (a) repetitive engagement in the behavior despite evidence of associated impairment; (b) personality correlates such as neuroticism; and (c) affective characteristics, such as high levels of negative affect—defined as a state of mind that can lead to avoidant, defensive, or hostile attitudes and behaviors [44]. Laghi et al. (2020) highlighted the role of emotional challenges and self-control in predicting binge and excessive alcohol consumption in young people [45]. In their study that included 849 students, they found that those at risk for alcohol abuse often exhibited low self-esteem, feelings of disconnection from themselves, difficulties managing emotions, and an extreme desire for self-control. Statistical analysis identified poor emotional regulation and rigid self-discipline as key predictors of binge and excessive drinking [45].

Given the significant relationship between binge behaviors and mental health challenges, the Operational Diagnosis of Mental Health and Addictions (2022), a public report by Mexico’s Ministry of Health, reported that the most common mental health disorders in Mexico include depression, affecting 5.3% of the population, and alcohol use disorder at 3.3% [46]. Mexico’s National Institute of Statistics and Geography (INEGI) reported that the national suicide rate increased from 6.2 per 100,000 inhabitants in 2020 to 6.8 in 2023, with Yucatán recording the country’s second-highest rate at 14.3 per 100,000 also in 2023 [46]. Despite the increasing prevalence of mood disorders, most Mexicans in need of mental health services do not seek professional support [30]. Data from the 2022 ENSANUT revealed concerning trends in alcohol use among young adults. Current alcohol consumption—defined as the percentage of participants who reported consuming at least one standard alcoholic drink in the past 12 months—was reported by 20.6%. Heavy drinking—defined as consuming five or more drinks on a single occasion for men and four or more for women—was reported by 13.9% in the past 12 months and 5.2% in the past 30 days [47].

#### 1.1.4. Screening

Early intervention, aimed at timely detection, screening, and diagnosis, generally has a positive impact on prognostic outcomes and overall disease burden [43]. Numerous ED instruments have been developed to identify symptoms or associated psychological traits. However, to date, none is a robust diagnostic marker or serves to detect risk factors or the likelihood of developing an ED [10]. There is a dearth of information from middle-income countries, and the higher prevalence of ED in Mexico accompanied by lifestyle changes among young adults that include diet, exercise and the new global beauty standards brought by technology (TV advertisement and beauty magazines) were exacerbated by the concerning mental health indicators post confinement, that reported an increase in anxiety and depression symptoms among young adults [48]. Thus, it was decided to field a study with the aim of analyzing the frequency of significant ED symptomatology and its correlation with a sample of students seeking mental health services at one of the participating universities (UNAM) and students from another university (UADY) that sampled undergraduate students not seeking mental health services. Gender, sexual orientation, BD, symptoms of anxiety and depression, and patterns of alcohol consumption were included. The study aim was to shed light on these associations due to their high prevalence and adverse health consequences associated with ED and related behaviors. Early screening and characterizing risk factors can inform targeted interventions that could mitigate mental health outcomes and reduce the burden of disease among college students in a middle-income country and worldwide.

## 2. Materials and Methods

### 2.1. Study Design

According to Feinstein’s classification, the study is an observational (non-intervention); cross-sectional (one measurement); and comparative study (two samples) [49].

### 2.2. Sample

The study population included college students from two participating institutions (public universities), aged 18 to 30, that were approached and asked to participate during July 2024 and March 2025:Students attending the Faculty of Medicine at Yucatan’s Autonomous University (UADY) comprised by undergraduate students enrolled in health-related fields (nutrition and rehab medicine) who did not seek mental health services.Students from several disciplines enrolled at the National Autonomous University of Mexico (UNAM) who sought mental health services at the Department of Psychiatry and Mental Health (DPSM).

The sample size was calculated using EPIDAT 4.3 with the formula for confidence intervals for a proportion, considering an infinite population for students enrolled at UADY and a finite population for students seeking mental health services at UNAM. A 95% confidence level was used, with a precision of 0.5% for UADY and 1% for UNAM, resulting in a minimum sample of 473 students from UADY and 153 students from UNAM. A non-probabilistic quota sampling method was used because accessing the entire population of students seeking mental health services at UNAM was not feasible, making random selection impossible. Similarly, all UADY students that were approached and agreed to participate were included in the survey and due to the lack of a defined sampling frame this did not allow for probabilistic selection procedure. This strategy maximized participation given the logistical and structural conditions of each university.

#### Selection Criteria

Inclusion: Students who sought care at the DPSM/UNAM mental health clinics (mostly due to depression and anxiety disorders), and students enrolled in health sciences and nutrition at Yucatan’s UADY; aged 18 to 30 years; who agreed to participate in the study and signed the informed consent form.

Exclusion: Students presenting with psychotic symptoms at UNAM’s mental health services; students requesting a crisis intervention; and students presenting with acute suicide risk.

Elimination: Students who did not complete the screening instruments adequately; students whose responses had more than 20% of the survey questions left unanswered, in the online or paper-and-pencil format; students who no longer wished to participate in the study.

### 2.3. Data Collection Instruments

#### 2.3.1. Demographic Characteristics of Participants

The demographic variables included age, sex (assigned at birth), gender identity, sexual orientation and major enrolled in (medicine, health related sciences and others).

#### 2.3.2. ED Symptomatology

Binge Eating Scale (BES). The BES (Gormally et al. 1982) assesses severity of binge eating, including behavioral, emotional, and cognitive manifestations [50]. It includes 16 items: eight behavioral (e.g., rapid eating, large food intake) and eight emotional/cognitive (e.g., fear of not being able to stop eating) [50]. The BES was rated according to standard guidelines, with items ranging from 0 to 3. A total score ≥17 was used as the cut-off to classify participants based on the scale’s standardized criteria. This study used the version adapted and validated in Mexico by Zúñiga and Robles (2006) and recently validated by Valdez-Aguilar (2022) [51,52]. The reliability of this instrument in Mexico was α = 0.90 [53]. The reliability for this study’s sample was similar (ω = 90).

Eating Attitudes Test—26 Items (EAT-26). The EAT-26 is a screening tool used to identify ED risk based on attitudes, emotions, and behaviors related to food. The instrument includes three subscales (dieting, bulimia/food preoccupation, and oral control), with total scores ranging from 0 to 70. A total score ≥20 was used as the cut-off following standard guidelines [54,55]. This study used the version validated in Mexico (Lugo-Salazar, 2019) with a reliability score among Mexican young adults of α = 0.82 [56], and across this study’s sample of ω = 85.

#### 2.3.3. Body Dissatisfaction (BD) by Sex

Body Shape Questionnaire (BSQ)—Women. To assess BD among female participants, the BSQ was created by Cooper et al. (1987) [57]. The original BSQ is a 34-item self-report inventory measuring general concerns about body image, specifically the subjective experience of “feeling fat”. The BSQ has been used to assess body weight and shape concerns among patients with BD [5,58]. In this study we used the abridged version [59]. Students responded based on how they felt about their body shape over the past four weeks. Items are scored on a 6-point Likert scale (1 = never to 6 = always), with a total score range from 34 to 204. In general, a score ≥110 indicates BD [59]. The BSQ has been widely validated in Mexico by Vázquez Arévalo (2011) and shows high internal consistency (α = 0.93–0.98) [59]. Mean scores reported are 71.9 (SD = 23.6) among UADY students and 136.9 (SD = 22.5) among people diagnosed with bulimia nervosa [57,59]. The reliability of BSQ for this study’s sample was ω = 97.

Drive for Muscularity Scale (DMS)—Men. The DMS was developed by McCreary et al. (2004) and originally validated among Canadian men [60]. While women often express BD related to abdominal fat, men typically present muscularity dissatisfaction [61]. They aspire to a muscular physique which in itself constitutes a socially reinforced ideal [61,62]. The instrument includes 15 items scored on a 6-point Likert scale, where 0 = always, and 6 = never [62]. Escoto et al. (2013) conducted an exploratory factor analysis among 369 Mexican male college students (mean age = 21) and identified reliability of α = 0.86 [62]. The reliability for the DMS among this study’s sample was ω = 90.

A dichotomous variable, BD, was created based on scores from the BSQ and DMS scales to capture a broader conceptualization of body image concerns, integrating both shape dissatisfaction [63] (assessed with the BSQ among women) and preoccupation with increasing muscularity [64] (assessed with the DMS among men). Because the two instruments differ in their scoring systems and number of items- and only the BSQ has a validated cut-off score for Mexican adults- a median split was used to homogenize both measures and categorize them in two groups: high BD and low BD. High BD was defined as scoring above the median on either measure in our sample (BSQ ≥ 85 for women or DMS ≥ 40 for men). This median-split approach has been employed in previous research on BD [65,66].

To examine whether the association between BD and ED symptomatology was robust across different BSQ thresholds, logistic regressions were performed using both the ≥85 and ≥110 cut-off scores. Both models were statistically significantly, with participants above either threshold showing substantially higher odds of exhibiting significant ED symptomatology. The model for the ≥85 threshold was significant (χ^2^ (1) = 118.61, *p* < 0.001, OR = 39.38, Nagelkerke R^2^ = 0.40). When applying the ≥110 cut-off, the results remained strong (χ^2^ (1) = 143.43, *p* < 0.001, OR = 26.34, Nagelkerke R^2^ = 0.47).

#### 2.3.4. Symptoms of Depression and Anxiety

Hospital Anxiety and Depression Scale (HADS). Developed by Zigmond and Snaith (1983), the HADS identifies possible/probable cases of anxiety and depression in non-psychiatric settings [67]. Items reflecting somatic symptoms (e.g., dizziness, headache, insomnia) or severe mental disorders were excluded to focus on typical anxiety/depression features. The HADS has been used in Spain [68] and in Mexican clinical populations [69,70]. Reliability of this instrument as per data from Mexico was α = 0.90, indicating high internal consistency [71]. The reliability among this study’s sample was ω = 0.74.

#### 2.3.5. Alcohol Consumption

Alcohol Use Disorder Identification Test (AUDIT). The AUDIT was used to assess alcohol consumption due to the known comorbidity between BED alcohol use. Higher scores indicate greater alcohol consumption [72]. The scale has been validated in Mexico with a reliability of α = 0.80 in young adults [73,74,75] and ω = 0.78 in the present sample.

### 2.4. Data Collection Procedure

A preliminary four-week period fielded in both universities included drafting and posting fact sheets on the most common ED (BN, AN and BED) and their association with the most common mental health disorders. A different ED was described in leaflets in common areas of the campus and in week four an invitation was posted throughout the campuses (UNAM and UADY) asking students to participate in a research project.Students enrolled in medicine and health related sciences (at Yucatan’s UADY) and those seeking mental health services (at UNAM) were recruited based on inclusion criteria (see Figure 1)Eligible students received a full explanation of the study and its objective, including the association between ED and mental health. Informed consent was obtained.After signing informed consent, students completed five screening tools: all participants completed the EAT-26, BES, AUDIT, and HADS; females additionally completed the BSQ, and males the DMS. Participants were accompanied by a member of the study team who was available to address queries and ensure proper completion of the study instruments.A digital version of all screening instruments as well as paper pencil surveys were used for students who did not have good internet access or preferred this format.Sociodemographic and clinical data were collected: age, sex assigned at birth, gender identity, sexual orientation, major (medicine, health related sciences and others), symptoms of anxiety or depression, alcohol consumption patterns, symptoms of BD and ED symptomatology.After completing the instruments, which were fielded at UADY and then stored at UNAM’S central database, we analyzed the responses of all six screening tools. Paper and pencil surveys mostly fielded at UNAM were entered in the central database that included UADY and UNAM students.Participants who requested their results received them via email.All data was stored for analysis and was accessible only to members of the research team. Students were assigned alphanumeric codes to ensure anonymity.

### 2.5. Data Analysis

Data analysis was done using IBM Corporation, Armonk, New York, USA—IBM Statistical Package for Social Sciences (SPSS), Version 23 [76]—and Python Software Foundation, Wilmington, Delaware, USA—Python 3.11 and Stats models 0.14.4 [77]—to carry out statistical analysis. Measures of BD, symptoms of depression and anxiety and alcohol consumption were described through means and standard deviations (SD). Significant ED symptomatology was described with frequencies and percentages.

For continuous scale scores, normality was tested using the Shapiro–Wilk test, and homogeneity of variances was assessed using Levene’s test (alpha = 0.05). When both groups were approximately normally distributed, they were compared using Welch’s *t*-test (two-sided), which does not assume equal variances. When normality was not met or distributions were clearly skewed or prone to outliers, the Mann–Whitney U test was used.

For binary or categorical variables, associations between UNAM and UADY samples (e.g., presence of significant ED symptomatology) were assessed using contingency-table analysis. When all expected cell counts exceeded five (Cochran’s criterion), Pearson chi-square test of independence was applied. When any expected frequency was below five, Fisher’s exact test was used, as it provides an exact *p*-value without relying on large-sample approximations. These tests assess whether category membership in one variable is statistically associated with that in another, rather than comparing group means as in the continuous-scale analyses.

Because participants completed the survey using either the paper-and-pencil or online formats, a Mann–Whitney analysis was conducted to compare responses between formats. The results showed no significant differences, suggesting consistency across data collection methods.

All analyses were carried out using two-tailed tests, with *p* ≤ 0.05 serving as the criterion for statistical significance. For the DMS, BSQ, EAT-26, BES, AUDIT and HADS scales, normality was rejected by the Shapiro–Wilk test at *p* ≤ 0.05.

## 3. Results

### 3.1. Demographic Characteristics of Participants

A total of 846 participants responded to the online format of the survey compared to 53 who answered the pencil and paper format. From the original sample, 289 students were not included in the analysis, mostly due to incomplete responses (92.7%) or because they did not meet the age criteria (7.3%). Most of the participants who did not complete the surveys had responded to the online format (97.9%) rather than via the paper and pencil format (2.1%). The final sample included 610 students who completed five screening tools. All final respondents completed the EAT-26, BES, HADS, and AUDIT scales. Of the 610 participants 379—mostly cisgender women—completed the BSQ, while 231—mostly cisgender men—completed the DMS. From the total sample, 457 (74.9%) were enrolled at UADY and 153 (25.1%) at UNAM.

Participant’s mean age was 21 years (SD = 2.9). Table 1 summarizes the characteristics from both UADY and UNAM samples. Significant differences were found in sex assigned at birth (*p* < 0.001), gender identity (*p* = 0.002), sexual orientation (*p* < 0.001), and major enrolled in (*p* < 0.001). The UNAM sample included more non-cisgender (*p* = 0.002) and non-heterosexual (*p* < 0.001) participants and was composed mainly of medical students.

### 3.2. Screening Surveys

#### 3.2.1. Significant ED Symptomatology

When merging the UADY and UNAM samples, 19.8% of students displayed significant ED symptomatology, as assessed by the BES and EAT-26 scales. The UNAM sample showed higher scores on both measures (BES: 11.03 ± 8.41 vs. 8.02 ± 6.67, *p* < 0.001; EAT-26: 9.37 ± 9.96 vs. 6.99 ± 7.44, *p* = 0.009) compared to UADY. Table 2 presents the distribution of significant ED symptomatology across demographic groups, with corresponding odds ratios (OR) and 95% confidence intervals (CI).

A simple logistic regression analysis including HADS and AUDIT scores was conducted to examine their direct association with significant ED symptomatology. Anxiety and depressive symptoms (assessed with HADS) showed a positive and statistically significant effect (B = 0.15, SE = 0.02, OR = 1.16 *p* < 0.001). Alcohol consumption patterns showed a smaller but significant association (B = 0.07, SE = 0.03, OR = 1.08, *p* = 0.002).

Logistic regressions were conducted separately for the DMS and BSQ to validate their equivalence and to create the BD variable. Both measures of BD significantly predicted ED symptomatology with *p* < 0.05, although the model including BSQ accounted for a larger proportion of variance (Nagelkerke R^2^ = 0.460) compared to the DMS model (Nagelkerke R^2^ = 0.252).

A multiple logistic regression model was performed to analyze variables that affected significant ED symptomatology. The model explained a substantial portion of the variance in significant ED symptomatology (Nagelkerke R^2^ = 0.39), supporting the relevance of the included predictor (see Table 3). The strongest predictor was the presence of high BD among students (*p* < 0.001).

Table 2 presents the results of a multiple logistic regression model examining predictors of significant ED symptomatology. The analysis indicated that cisgender men were significantly less likely than cisgender women to exhibit significant ED symptomatology (OR = 0.30, 95% CI [0.17–0.53], *p* < 0.001). High BD emerged as the strongest predictor (OR = 13.46, 95% CI [6.79–26.26], *p* < 0.001), and higher anxiety and depression symptoms (measured with HADS) were also associated with an increased likelihood of displaying significant ED symptomatology (OR = 1.10, 95% CI [1.04–1.16], *p* < 0.001). Other variables, including major enrolled in, sexual orientation, and alcohol consumption were not significant predictors.

The total sample was separated into quartiles of BD to analyze the association between BD and significant ED symptomatology. As shown in Figure 2, the frequency of significant ED symptomatology increased progressively across BD quartiles: 1.04% in Q1, 4.12% in Q2, 15.73% in Q3, and 36.97% in Q4.

#### 3.2.2. Symptoms of BD

Over half of the total sample reported high levels of BD (50.4%). Nearly half of UADY students (48.4%) and over half of UNAM students (56.2%) exhibited significant symptoms of BD; however, this difference was not significant (*p* 0.079).

#### 3.2.3. Symptoms of Depression and Anxiety

A Mann–Whitney U test was conducted to compare HADS scores between UNAM and UADY samples. Results indicated a statistically significant difference between the two groups, U =18181, *p* < 0.001. Students from UNAM (Median = 20.0, M = 18.97, SD = 5.10) reported significantly higher HADS scores than students from UADY (Median = 14.0, M = 14.60, SD = 4.75).

#### 3.2.4. Alcohol Consumption

The alcohol consumption pattern was statistically higher among UNAM students with an AUDIT mean of 3.25 (SD= 4.39, 95% CI 2.55–3.95) compared to UADY’s mean of 1.86 (SD = 3.03, 95% CI 1.59–2.13; *p* < 0.001). The mean score across both UNAM and UADY was 2.21 (SD = 3.47, 95% CI 1.94–2.49).

## 4. Discussion

This study aimed to identify the frequency and factors associated with significant ED symptomatology in undergraduate students at two public universities in Mexico. A comparison was conducted between students seeking mental health services at UNAM, Mexico City’s largest university [78], and students enrolled in health-related majors at UADY, the most prestigious public university in Yucatán. Yucatán was selected as a comparison state because of its mental health indicators [20,79]. Given that UNAM students were actively seeking mental health services, this group was expected to exhibit higher ED symptomatology and more adverse mental health indicators than students attending UADY. The study also aimed to document the associations between significant ED symptomatology, BD, anxiety and depression symptoms, and alcohol consumption patterns [80,81]. As anticipated, UNAM students scored higher across all screening instruments, enhancing the contrast between the two groups. To our knowledge, this is the first study to simultaneously assess significant ED symptomatology and the emerging prevalence of BD, anxiety, depressive symptoms, and alcohol consumption patterns among young adults enrolled in public universities in Mexico.

The present study found a prevalence of 19.8% of significant ED symptomatology (UNAM: 32.7%; UADY: 15.5%). The highest frequency rates were observed among women, non-binary individuals, those with high BD, and those reporting elevated anxiety and depressive symptoms. These findings align with previous research showing that ED prevalence is higher among women than men [21]. Moreover, the results highlight a recently overlooked issue within LGBT+ communities: while cisgender individuals are typically overrepresented in the literature, non-cisgender individuals remain understudied despite exhibiting high rates of ED risk [82]. Beyond these demographic patterns, mental health emerges as a central factor in understanding ED risk. Prior studies have indicated that more than half of individuals with ED present comorbid anxiety or depressive conditions [41], and this pattern is reflected in our sample. Notably, the UNAM sample —composed of students actively seeking mental health services at the DPSM, which frequently treats mood disorders—reported both higher HADS scores and higher ED symptomatology compared with the UADY sample, which included students not actively seeking mental health services. However, depression and anxiety symptoms were also noted among UADY’s Yucatán sample and are consistent with prior state and national data, where Yucatan ranked second in suicidality in 2023 [83], a situation further exacerbated following the COVID-19 pandemic [84]. These contextual factors may help explain why significant ED symptomatology in the Yucatan sample reached 15.5%, compared with 9% in women and 13% in men reported in 2008 [19]. Overall, these results underscore the central role of mental health in increasing significant ED symptomatology among Mexican college students, including individuals who do not actively seek mental health services.

The study’s overall ED frequency aligns with González-Alvarado (2024), who reported an ED prevalence among college students of 37.3% in women and 32.6% in men, as well as with government estimates that recently reported an ED symptomatology prevalence of 25% [17]. These findings suggest that a significant proportion of Mexican college students are at risk of developing an ED, with higher symptomatology among UNAM students likely reflecting the underlying mood disorder profile of this group. Overall, significant ED symptomatology among Mexican students falls between previously reported national and global prevalence estimates, underscoring their significance as a public health concern in Mexican universities.

Approximately half of all students from both universities displayed significant levels of BD. Students seeking mental health services at UNAM scored higher than their UADY counterparts, and BD emerged as the strongest predictor of significant ED symptomatology, increasing the odds by 13.46-fold. Consistent with prior evidence linking BD to gender identity [85], significant ED symptomatology across gender was also examined. As reported by Giel et al. (2022) [86], cisgender women exhibited statistically significantly higher levels of ED symptomatology compared to cisgender men. These findings are consistent with previous research [24], in which women tend to have significantly more obsession with thinness and BD [87]. Two decades ago, before the advent of widespread social media use, the prevalence of BD in Mexico stood at 16% among women and 22% among men [19]. By 2020, 61% of adolescents reported BD [88], reflecting social media’s pervasive influence [89] confirmed by the over five hours per day that Mexico’s youth currently spend on social media [90]. The high frequency of BD in this sample underscores its role as a central risk factor for reporting significant ED symptomatology and emphasizes its relevance as a public health concern among university students. The markedly higher risk among non-cisgender students indicates a particularly vulnerable group that requires closer attention in future research, ideally using larger samples. Together, these results suggest that sociocultural pressures and digital media exposure are shaping body image concerns in Mexico, contributing to ED development in both UNAM and UADY samples.

Mexico City’s UNAM students exhibited higher alcohol consumption than their Yucatan’s UADY peers. Alcohol use was associated with significant ED symptomatology in the bivariate analysis, but this association disappeared in the multivariate analysis. The link between alcohol consumption patterns and ED symptomatology can be explained by emotional dysregulation, which is considered an etiological factor for BED and alcohol use disorder [91]. The AUDIT scores in this study contrast with Mexico’s 2017 National Survey on Alcohol and Tobacco (ENCODAT), which reported higher alcohol use in Yucatan compared with the national average [92], and with previous studies suggesting an association between alcohol consumption and ED [93]. Nevertheless, it is important to note the characteristics of the UNAM sample, which included students actively seeking mental health services, potentially reflecting the well-established link between alcohol consumption and mental health disorders [94]. The lack of a statistical association with significant ED symptomatology in the multivariate analysis suggests that alcohol may play a secondary role compared with other factors, such as anxiety symptoms and high rates of BD.

Several previously validated screening instruments were included, supporting the reliability of the collected data. Nevertheless, some limitations should be acknowledged. The cross-sectional design prevents any inference of causality. Participants were recruited through convenience sampling. Some were seeking mental health services at UNAM, while all participants came from diverse social contexts. Consequently, the UNAM and UADY study groups may differ in sociocultural background, potentially introducing confounding bias related to mental health outcomes. Additionally, the UADY sample was not representative of the broader UADY student body. Self-reported measures of sensitive behaviors may also be biased, as indirect methods often yield more accurate estimates [95,96,97]. Participants who completed the online survey exhibited more missing data than those completing the paper and pencil version. Ceccato et al. (2024) noted that digital surveys often experience a significant drop-off after the initial questions, whereas the paper surveys show a more stable pattern of responses, which could explain this discrepancy [98].

In addition, although the logistic regression models showed statistically significantly effects for both DMS and BSQ, the explanatory power of the DMS model was limited. The BSQ scores accounted for a larger proportion of the variance, consistent with prior evidence linking body image concerns and the presence of ED behaviors. Although DMS showed a statistically significant association, the effect size was relatively weak (R^2^ = 0.252). This suggests that, within our sample, drive for muscularity explains only a modest proportion of the variance in men’s ED symptomatology. While the DMS can assess muscularity-oriented attitudes and behaviors associated with higher BD and ED risk [99], it does not assess the broader pathological features of muscle dysmorphia, which are better captured by the Muscle Dysmorphic Disorder Inventory [100]. This distinction helps explain the modest predictive power observed: muscularity drive alone does not encompass the cognitive preoccupation, distress, and functional impairment associated with more ED symptomatology. In this study, the composite BD variable was designed to capture a broader conceptualization of body image concerns, integrating both shape-related dissatisfaction and muscularity-related dissatisfaction.

BMI data were not recorded at either university, which represents a limitation because BMI is a key factor in assessing ED symptomatology severity [101,102]. Furthermore, the small number of non-cisgender and “other” gender identity students enrolled in certain disciplines limited the ability to analyze these subgroups in greater depth. Finally, private universities could not be surveyed; however, these institutions represent a smaller proportion (37.9%) of the university student population in Mexico [103], which likely limits the impact of their exclusion on the overall findings.

## 5. Conclusions

This study found that nearly one in five Mexican college students from two public universities exhibited significant ED symptomatology, with higher rates among those seeking mental health services at UNAM compared to students from UADY. The highest frequencies were observed among women and non-cisgender individuals. High BD emerged as the strongest factor associated with significant ED symptomatology, followed by symptoms of anxiety and depression. In contrast, alcohol consumption patterns were not statistically associated after adjustment. These findings underscore the importance of early screening for mental health conditions, BD, and ED among young adults in Mexico, as well as the need for targeted prevention and intervention strategies in university settings.

## Figures and Tables

**Figure 1 ijerph-22-01797-f001:**
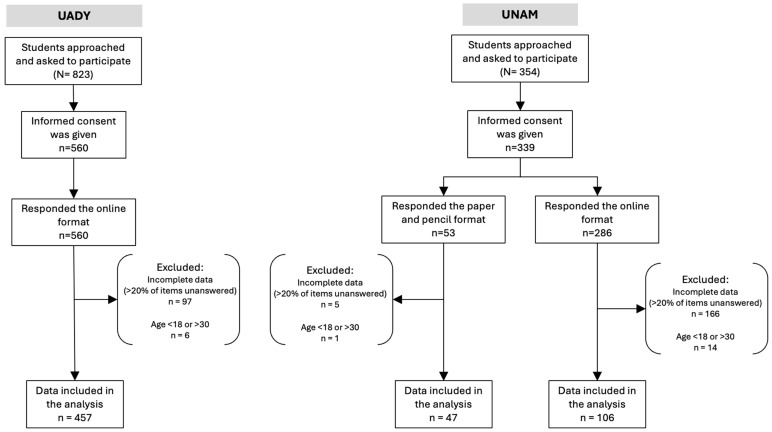
Sample selection flowchart. Note. >20% of items unanswered include participant’s demographic characteristics and scores in the following screening instruments: BES, EAT-26, BSQ, DMS, HADS and AUDIT.

**Figure 2 ijerph-22-01797-f002:**
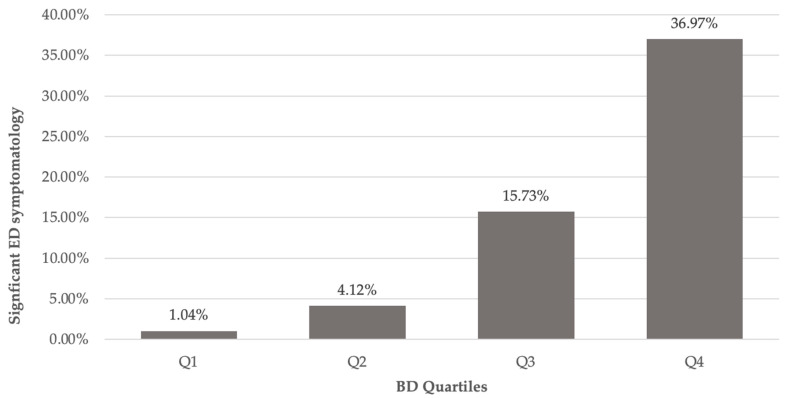
Significant ED symptomatology by BD quartiles. Note. BD was assessed using a dichotomic variable (DMS for men and BSQ for women). Significant ED symptomatology was assessed with the BES ≥17 and EAT-26 scales ≥20. Prevalence rates of ED symptomatology by BD quartile: Q1 (1.04%, N = 96), Q2 (4.12%, N = 97), Q3 (15.73%, N = 178), Q4 (36.97%, N = 238). Total sample size = 609.

**Table 1 ijerph-22-01797-t001:** Characteristics of participants compared by university.

Variable	UADY (n = 457)	UNAM (n = 153)	Statistic		Total (n = 610)
	*n (%)*	*n (%)*	*χ* ^2^	*p*	n (%)
Sex assigned at birth					
Man	172 (37.6)	58 (37.9)	0.001	<0.001	230 (37.7)
Woman	285 (62.4)	95 (62.1)			380 (62.3)
Gender identity					
Cisgender men	169 (37.0)	55 (35.9)	12.97	0.002	224 (36.7)
Cisgender women	281 (61.5)	87 (56.9)			368 (60.3)
Non-cisgender *	7 (1.5)	11 (7.2)			18 (3.0)
Sexual orientation					
Heterosexual	377 (82.5)	95 (62.1)	30.95	<0.001	472 (77.4)
Bisexual	58 (12.7)	39 (25.5)			97 (15.9)
Homosexual	19 (4.2)	11 (7.2)			30 (4.9)
Other	3 (0.7)	8 (5.2)			11 (1.8)
Major			264.3	<0.001	
Medicine	193 (42.2)	98 (64.1)			291 (47.7)
Nutrition	177 (38.7)	0 (0)			177 (29.0)
Physical Rehabilitation	87 (19.0)	0 (0)			87 (14.3)
Other majors *	0 (0)	55 (35.9)			55 (9.0)
Body Dissatisfaction (BD)			3.08	0.079	
Low BD	236 (51.6)	67 (43.8)			303 (49.7)
High BD	221 (48.4)	86 (56.2)			307 (50.3)
	M ± SD	M ± SD	*Z*	*p*	Total M ± SD
Age (years)	20.91 ± 2.57	22.07 ± 3.85	−3.21	<0.001	21.20 ±2.9
Symptoms of depression and anxiety (HADS)	14.60 ± 4.75	18.97 ±1.86	−8.81	<0.001	15.69 ±5.19
Alcohol consumption (AUDIT)	1.86 ±3.03	3.24 ±4.40	−3.50	<0.001	2.20 ±3.48
Body dissatisfaction—Women (BSQ)	87.54 ± 30.03	101 ± 39.09	−2.73	0.006	90.90 ±32.98
Body dissatisfaction-Men (DMS)	42.12 ± 14.97	40.98 ± 13.85	0.34	0.737	41.83 ±13.68
BED symptomatology (BES)	8.02 ± 6.67	11.03 ± 8.41	−3.86	<0.001	8.77 ± 7.25
High-risk eating behaviors (EAT-26)	6.99 ± 7.44	9.37 ± 9.96	−2.59	0.009	7.57 ± 8.19

Note. Values are based on the final sample of 610 participants, showing frequencies (n) and percentages (%), as well as means (M) and standard deviations (SD) for each university sample. BES = Binge Eating Scale; EAT-26 = Eating Attitudes Test–26; UNAM = Universidad Nacional Autónoma de México; UADY = Universidad Autónoma de Yucatán. * Non-cisgender includes non-binary, transgender men, “other” gender identity and students that “chose not to disclose”. * Other majors included students enrolled in Arts and Sciences.

**Table 2 ijerph-22-01797-t002:** Distribution of significant ED symptomatology and crude associations with participant’s characteristics.

Group		Significant ED Symptomatology			
		*n* *(%)*	OR	95% CI	*p*
College					
UADY	457	71 (15.5)	-	-	<0.001
UNAM	153	50 (32.7)	2.64	[1.73–4.03]	
Sex assigned at birth					
Men	230	27 (11.7)	-	-	<0.001
Women	380	121 (31.8)	2.47	[1.55–3.92]	
Gender identity					
Cisgender men	224	25 (11.2)	-	-	<0.001
Cisgender women	368	91 (24.7)	2.62	[1.62–4.22]	
Non-cisgender *	18	5 (27.8)	3.06	[1.01–9.30]	
Sexual orientation					
Heterosexual	472	82 (17.4)	-	-	<0.001
Bisexual	97	32 (32.9)	2.34	[1.44–3.81]	
Homosexual	30	5 (16.7)	0.95	[0.35–2.56]	
Others	11	3 (27.3)	1.78	[0.46–6.84]	
Major					
Medicine	291	63 (21.6)	-	-	0.029
Nutrition	177	24 (13.6)	0.58	[0.34–0.99]	
Physical Rehabilitation	87	17 (19.5)	0.87	[0.46–1.64]	
Other majors *	55	17 (30.9)	1.68	[0.87–3.25]	
Body Dissatisfaction-Women (BSQ)					
Low BD	101	11 (3.6)	-	-	<0.001
High BD	104	90 (46.4)	3.68	[2.64–4.70]	
Body Dissatisfaction-Men (DMS)					
Low BD	182	4 (2.2)	-	-	0.005
High BD	86	19 (18.1)	1.31	[0.35–2.28]	

Note. All 95% confidence intervals were calculated using the Wilson score method. Groups are only shown when they include at least five participants with significant ED symptomatology and five participants without significant ED symptomatology ($n\times \hat{p}\geq 5\;$ and $n\times (1-\hat{p})\geq 5$). Baseline groups are marked with “–” to indicate the reference for odds ratio comparisons. * Non-cisgender includes non-binary, transgender men, “other” gender identity and students that “chose not to disclose”. * Other majors included students enrolled in Arts and Sciences.

**Table 3 ijerph-22-01797-t003:** Multiple logistic regression analysis for significant ED symptomatology.

	B	S.E.	Wald	OR	95% CI	*p*
College						0.174
UADY (ref)	0.00	-	-	-	-	-
UNAM	0.49	0.36	1.85	1.64	[0.80–3.35]	0.174
Major						0.542
Medicine (ref)	0.00	-	-	-	-	-
Nutrition	−0.46	0.34	1.86	0.63	[0.33–1.21]	0.172
Physical Rehabilitation	−0.11	0.39	0.09	0.89	[0.41–1.92]	0.759
Others	−0.21	0.45	0.20	0.82	[0.33–1.97]	0.651
Gender identity						<0.001
Cisgender women (ref)	0.00	-	-	-	-	-
Cisgender men	−1.196	0.29	17.0	0.30	[0.17–0.53]	<0.001
Non-cisgender *	−0.77	0.63	1.47	0.47	[0.13–1.64]	0.226
Sexual orientation						0.490
Heterosexual (ref)	0.00	-	-	-	-	-
Bisexual	0.41	0.30	1.81	1.50	[0.83–2.71]	0.179
Homosexual	0.27	0.59	0.21	1.31	[0.41–4.18]	0.614
Others **	−0.51	0.99	0.27	0.60	[0.08–4.56]	0.604
Body dissatisfaction						<0.001
Low BD (ref)	0.00	-	-	-	-	-
High BD	2.60	0.34	56.94	13.46	[6.79–26.26]	<0.001
Symptoms of anxiety and depression (HADS)	0.09	0.03	12.65	1.10	[1.04–1.16]	<0.001
Alcohol consumption (AUDIT)	0.05	0.03	2.89	1.06	[0.99–1.12]	0.089
Constant	−4.59	0.54	72.21	0.01	-	<0.001

Note. OR = odds ratio; CI = confidence interval; ref = reference group; HADS = Hospital Anxiety and Depression Scaley; AUDIT = Alcohol Use Disorders Identification Test. * Includes non-binary, transgender men, chose not to disclose and “another”. ** Includes demisexual, asexual, queer and pansexual.

## Data Availability

The original contributions presented in this study are included in the article. Further inquiries can be directed to the corresponding author.

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
