# Peer review of "Frequency, Correlates, and Symptom Severity of Eating Disorders Among College Students in Mexico"

_ijerph, 2025, doi:10.3390/ijerph22121797_

Round 1

Reviewer 1 Report

Comments and Suggestions for Authors

The paper is generally well-written and its findings are promising. However, there are several significant methodological issues that need to be reconsidered:

  1. In the method section, there is no clear statement on which of the 266 individuals were excluded and why. What exactly does the phrase “mostly students who responded to online vs paper and pencil format” (p. 7, L 297-298) mean? Were the 266 excluded because they had incomplete surveys and were they only those who completed the survey online? Of the 266, does "mostly" mean that more than half were online respondents, but isn’t the exact percentage given? Was there any kind of selection bias, did online users have more incomplete surveys? In lines 480-481 it is mentioned: “A higher number of participants who completed the online version of the survey had more missing data, compared to those who completed the paper and pencil format. Ceccato et al. (2024) explained that digital surveys show a significant drop after the initial questions, while the paper surveys show a more stable pattern of responses, which could explain the difference we found.” This means that online respondents had a higher rate of incomplete surveys, and paper respondents had a lower rate of incomplete surveys, but this was not clearly stated in the Methods section. It is suggested that the following questions be answered and mentioned:
  • How many of the 266 were online respondents? (It says “mostly” meaning >50%, but no exact percentage is given).
  • What was the percentage of incomplete surveys per format (online/paper)?
  • If the criterion was >20% missing data, was it applied equally to both online and paper?
  • How many of the 876 submitted online vs paper?

All this significantly impacts the results because: If most of the 266 excluded were from UADY, then the UADY sample is smaller than the 457 stated, and the UNAM sample is closer to 153, creating an asymmetrical sample. If online respondents had higher missing data rates, and if more women used the online format, then women in the final sample are “overrepresented” compared to men, and this could explain why the study finds very high ED rates in women.

  1. In the abstract (L 45-47), the CI [0.99-1.12] does not align with OR=13.35 and suggests non-significance. Also, the CI is different from Table 2. Please check and correct.
  2. Different cut-offs are applied with insufficient justification, and it is unclear whether the combination of BSQ/DMS is psychometrically correct. Clearly specify the methods used to create the BD variable and provide bibliographic justification for the cut-offs.
  3. The authors combined two fundamentally different gender-specific measures, the Body Shape Questionnaire measuring body shape concerns and the Drive for Muscularity Scale measuring muscularity desires, into a single dichotomous BD variable using median splits rather than established clinical thresholds. While mathematically correct (379+231=610), this composite approach presents a critical validity concern for the study's primary finding (OR=13.35 for High BD predicting ED). Specific concerns: The BSQ and DMS measure distinct psychological constructs (shape anxiety vs muscularity drive), but no evidence is provided that these constructs have equivalent predictive relationships with ED symptomatology. The use of median splits deviates from clinical standards (BSQ clinical cut-off is ≥110, not ≥85), potentially inflating High BD prevalence and artificially strengthening the BD-ED association. Without gender-stratified analyses or sensitivity testing, the reported OR=13.35 cannot be evaluated as a unitary finding across both constructs. It suggested to: a) Provide separate logistic regression models for BSQ (women) and DMS (men) predicting ED symptomatology to determine if effects are equivalent, b) Conduct a sensitivity analysis using the established clinical BSQ cut-off (≥110) to assess whether findings remain strong when using standard thresholds, c) Clarify in Methods and Discussion sections that the composite BD variable represents a broader conceptualization of body image concerns encompassing both shape and muscularity dissatisfaction, mentioning potential differences in their mechanisms relative to ED risk, d) Consider z-score standardization if combining constructs.
  4. Line 247: The Cronbach's α=0.63 (HADS-A) and α=0.54 (HADS-D) are low, considered inadequate, and give doubt on the results. It is suggested that the authors recalculate the reliability or comment on the low value in the Limitations section, and consider the possibility of alternative measurement or data security adjustments.
  5. BMI is a key factor in eating disorders. The lack of recording prevents the evaluation of organic parameters, reducing the ability to control for confounders. It is suggested that the absence of this data (BMI) be included in the Limitations section, and if possible, a plan for future data collection be mentioned, or the methodological impact be mentioned.
  6. Figure 1 does not display the complete quartile distribution (Q2/Q3 are missing in L360-370) and the correlation claim is not properly presented. Also, some CIs in the tables include 1.0 without commenting on non-significance. It is suggested that the figure details be completed, the statistical tests supporting the correlations be commented on, and a clear distinction be made between significant and non-significant findings in the tables.
  7. Missing footnotes in Figures and Tables.
  8. Basic descriptive statistics for EAT-26, BES, and missing data patterns are missing. Add means (Mean), standard deviation (SD), and frequency by major/university. Describe in detail the missing data patterns and the method used to address them.
  9. The quantification of depression in the multivariate analysis was not significant, although it was in the bivariate analysis. Possible collinearity with anxiety. It is suggested that the relationship between anxiety and depression be reassessed, that potential multicollinearity be mentioned, and that separate or combined analyses be proposed to avoid confounding.

The authors should particularly pay attention to the following:

  • The samples (clinical vs health) are not comparable.
  • The claim of “strong correlation” without correlation coefficients.
  • The Discussion section needs extensive revision to be consistent with the actual findings and to address the significant methodological weaknesses identified in the Methods and Results sections.
  • Introduction doesn't adequately set up why comparing clinical vs non-clinical samples is important. There is insufficient framing of expected differences between genders in ED risk factors. There is no discussion of social/cultural factors specific to Mexico. It is recommended to strengthen contextual background for Mexican population and provide more specific rationale for clinical/non-clinical comparison design.

Other suggested corrections:

  • Abstract. Double full stop (L 46)
  • Wrong abbreviation (L 61)?? “AN, BD, and BED primarily affect...” Do you mean BN instead of BD? 
  • Inconsistent p-value format. Sometimes: “p<.001” (line 334), sometimes: “p .001” (line 339). Please, choose one format and use it consistently.
  • Several stylistic and technical language issues require correction before publication. The writing contains instances of weak academic voice and unnecessary personalism. For example, the statement “Ours was a cross-sectional and comparative study" (Methods, L 159) is grammatically correct but stylistically weak for academic publication. Similar stylistic improvements are needed throughout, where weak constructions (“was”) could be replaced with stronger action verbs (“conducted,” “examined,” “compared” etc).

The research has value and can contribute to the literature on eating disorders, but in order to be reconsidered for publication, major revisions must be made.

Reviewer 2 Report

Comments and Suggestions for Authors

It might be useful to specify that this is a ‘cross-sectional comparative study’.

In the abstract, it would be better to explain the use of OR: ‘OR=13.35; 95% CI 6.79–26.26’, since in the text the range reported is: ‘0.99–1.12’. Is this a typo?

What are the specific new findings of the study?

I believe it would be useful to explain the reason for using non-probabilistic quota sampling.

In Tables 1 and 2, standardise the precision of the OR and CI values.

What is the overall Nagelkerke R² or pseudo-R² value of the model?

Reviewer 3 Report

Comments and Suggestions for Authors

Comments are attached.

Round 2

Reviewer 1 Report

Comments and Suggestions for Authors

Thank you for the extensive revisions in response to the first review round. The paper has been meaningful revised. The authors revised the Methods and Results sections, corrected statistical errors, added missing descriptive data, conducted additional analyses (including analyses and gender-specific logistic regressions), clarified the BD variable, revised the Introduction, corrected stylistic and formatting issues, and expanded the Limitations section. The authors have shown careful attention to the corrections in the first review round and have successfully improved the manuscript. However, the current revised manuscript shows that some points are not fully handled and require targeted additions such as:

  1. The Introduction does not explain why gender differences are expected in Mexico. The Introduction mentions "Recent studies suggest that ED in men may be underdiagnosed," but does not explain the mechanistic “why” (i.e. that men face pressure toward muscularity/strength while women face appearance/thinness pressures, and LGBTQ+ populations face compounded vulnerabilities.) I suggest after line 107 to add approximately 100-120 words with references to address those gaps. The Introduction would benefit from systematically connecting gender, cultural context, and LGBTQ+ vulnerabilities and social media.
  2. Also missing in Introduction: Yucatan mental health burden (already mentioned in Discussion but it is suggested to add it in introduction too). So, after line 108 add a brief paragraph to include the above information establishing why Yucatan was selected and its relevance to the research rationale.
  3. It is recommended to expand briefly Discussion section about DMS Weak Effect (approx. 100-120 words). The current Discussion identifies the DMS weak effect but doesn't explain why it occurs or what it means for understanding men's ED vulnerability. The fact (DMS R²=0.252 is weak) is stated but does not explain why or what it means. Alternative etiological factors for men's ED are not discussed. Implications (clinical) for interpretation of findings aren’t explored. Mechanistic explanation for the weaker muscularity ED relationship is missing.
  4. Ensure uniform decimal formatting and alignment across Tables 2 and 3 (two or three decimals?).
  5. It would be helpful to state whether the BSQ≥85 and DMS≥40 thresholds used for the median split were derived from the present sample or based on prior literature.
  6. Add N per quartile in Figure 2 including sample sizes for each BD quartile.

Reviewer 3 Report

Comments and Suggestions for Authors

All suggested revisions have been incorporated, resulting in an improvement in the overall quality of the manuscript. The manuscript is now recommended for publication.

Author Response

We thank the Reviewer for the positive feedback and for recommending the manuscript for publication.